# Smoothies Reduce the “Bioaccessibility” of TiO_2_ (E 171) in the Model of the In Vitro Gastrointestinal Tract

**DOI:** 10.3390/nu14173503

**Published:** 2022-08-25

**Authors:** Ewa Baranowska-Wójcik, Dominik Szwajgier, Izabela Jośko, Bożena Pawlikowska-Pawlęga, Klaudia Gustaw

**Affiliations:** 1Department of Biotechnology, Microbiology and Human Nutrition, University of Life Sciences, Skromna Street 8, 20-704 Lublin, Poland; 2Institute of Plant Genetics, Breeding and Biotechnology, Faculty of Agrobioengineering, University of Life Sciences, Akademicka Street 13, 20-950 Lublin, Poland; 3Department of Functional Anatomy and Cytobiology, Faculty of Biology and Biotechnology, Institute of Biological Sciences, Maria Curie-Skłodowska University, Akademicka 19, 20-033 Lublin, Poland

**Keywords:** E171, TiO_2_, in vitro digestion, *Lactiplantibacillus plantarum B*, smoothie

## Abstract

The food colorant E171 (TiO_2_) containing nano fractions can cause potential health problems. In the presented work, we used a “gastrointestinal tract” model (oral→large intestine) to “digest” a fruit smoothie in the presence of TiO_2_ nanoparticles and the *Lactiplantibacillus plantarum* B strain. The TiO_2_ migration was measured using the microfiltration membrane (0.2 µm; model of “TiO_2_ bioacessability”). We observed that the addition of the smoothie reduced the Ti content in the microfiltrate (reduced “bioacessability”) at the “mouth”, “stomach” and “large intestine” stages, probably due to the entrapment of Ti by the smoothie components. A significant decrease in Ti “bioaccessibility” at the “gastric” stage may have resulted from the agglomeration of nanoparticles at a low pH. Additionally, the presence of bacterial cells reduced the “bioaccessibility” at the “large intestine” stage. Microscopic imaging (SEM) revealed clear morphological changes to the bacterial cells in the presence of TiO_2_ (altered topography, shrunk-deformed cells with collapsed walls due to leakage of the content, indentations). Additionally, TiO_2_ significantly reduced the growth of the tested bacteria. It can be stated that the interactions (most probably entrapment) of TiO_2_ in the food matrix can occur during the digestion. This can influence the physicochemical properties, bioavailability and in vivo effect of TiO_2_. Research aimed at understanding the interactions between TiO_2_ and food components is in progress.

## 1. Introduction

There is currently growing interest in nanoparticles. Their uniqueness stems from their physicochemical properties, which are increasingly incorporated in food production and pharmacochemical processes with a view to improving certain functional characteristics of products, e.g., their appearance, consistency, shelf-life, etc., [1,2,3]. E171 (TiO_2_) is a food additive found in a variety of food products (chewing gums, candies, blancmanges, sauces, cheeses, skimmed milk, ice-cream, cakes, dressings and certain powdered products). It is used to improve the color, brightness and “taste” of the products [4]. Food-grade TiO_2_ is a mixed-grade market product composed of micro particles (>100 nm) and nanoparticles (NPs), i.e., particles < 100 nm [5]. The latter fraction raised particular concerns in terms of its potential health impact [6]. When it was demonstrated that TiO_2_ NPs can be absorbed in the gastrointestinal tract, questions regarding their potential toxicity after chronic exposure began to arise [7]. The acceptable intake of TiO_2_ is considered to be 2.5 mg NPs/kg body weight/day [8]. However, it has been previously shown that, in some confectioneries, the concentration of E171 reached 2.5 mg Ti/g of food [9]. For example, the average TiO_2_ content in syrups, water-based ice and wine gums is, respectively, 0.28, 0.26 and 0.42 mg/kg of the product [9]. The age of the consumer, the dosage, the time of exposure and the route of absorption are factors that influence the toxicity of TiO_2_ NPs in humans [10]. It is estimated that a child may consume even 2–4 times more TiO_2_ than an adult when calculated in mg/kg b.w./day. For instance, 0.2–0.7 mg and 1 mg TiO_2_/kg b.w. are consumed per day in the USA and Great Britain, respectively. However, due to the higher consumption of sweets and the lower b.w., children under 10 years of age may ingest 1–3 mg TiO_2_/kg per day [8,10].

The gastrointestinal tract (GIT) is a link connecting the outside world with the organism’s internals, where nutrients are extracted from the ingested food [11]. The intestines, as a place where such nutrients are absorbed, are in constant contact with various additives used in food production [12]. Gastrointestinal disorders are complex by nature, but at this point, it has been conclusively shown that the use and consumption of food additives has indeed contributed to the increased incidence of gastrointestinal disorders such as inflammatory bowel disease (IBD) or irritable bowel syndrome (IBS) [13]. Due to their small size, NPs are able to penetrate the cellular barrier, causing oxidative stress and damaging the tract lining cells or the layer of mucus [14,15]. As follows from the latest reports, NPs may have adverse effects, contributing to the development of bowel and systemic inflammations, pathogenic changes to the composition of the intestinal microbiota [16,17] and the formation of precancerous lesions in the colon [18]. Numerous studies reported that the consumption of TiO_2_ NPs exerts negative effects including inflammation and damage to the liver, kidneys and heart [19,20], modifications of the cellular cycle and cellular membranes, apoptosis [21,22] or oxidative stress [23].

As food travels through the GIT, it undergoes specific structural and physicochemical changes [24]. Enzymes, digestive fluids and pH changes in sections of the tract can potentially influence in vivo absorption [25]. It has been demonstrated that TiO_2_ NPs can form complexes with polyphenols through the enediol functional groups [26], which, in turn, can affect the bioavailability of polyphenols [3]. Additionally, TiO_2_ NPs influence the growth of bacteria, as was previously shown in the case of selected strains of intestinal, lactic acid and opportunistic microorganisms, by adsorption and/or complex formation [10,27]. These interactions of bacteria and the food matrix with TiO_2_ can play a role by decreasing the “bioaccessibility” of TiO_2_ during digestion. The degree of degradation of food components increases the surface area and the specific volume of indigestible food polymers, predominantly of plant origin (dietary fiber). 

In vitro models of human intestines have been employed in many studies to facilitate the analysis of the effects of nutrient “bioaccessibility” and transport in the human organism [28,29]. In the present study, we determined whether a fruit smoothie can reduce the “bioaccessibility” of TiO_2_ (E 171) in an in vitro model of the gastrointestinal tract, especially in the presence of lactic acid bacteria. For this purpose, we simulated the digestion of a smoothie using an advanced in vitro model of GIT. The type of the product (smoothie) was chosen based on the advanced transformation of non-starch polysaccharides from the ingredients, due to which the formed 3D mesh can lower TiO_2_ NPs’ mobility within the food matrix, as verified in the course of the simulated digestion (by micro-filtering through a 0.2 µm membrane).

## 2. Materials and Methods

### 2.1. Preparation of TiO_2_ and the Smoothie for In Vitro Digestions

Food-grade TiO_2_ (E171) was purchased from a supplier in Poland: Food Colors, Reagan Str. 14, 97–300 Piotrków Trybunalski. A total of 1 g was suspended in 1 L of distilled water and sonicated for 30 min in a sonication bath filled with ice (Intersonic 101, Zakład Doświadczalny Podzepołów i Technologii Elektronicznych ITR, Warsaw, Poland, 100% power), followed by microfiltration (15 min, 0.2 µm, Vivaflow VF05P7 poly(ether)-sulfone (PES) membrane, Sartorius, France; MasterFlex L/S, Drive 900 peristaltic pump, Cole Palmer, Vernon Hills, IL, USA) with continuous sonication with ice (Sonic Vibra-Cell sonication head, Labo-Plus, Poland, 80% power). The TiO_2_ content in the microfiltrate was determined as described below (Section 2.4), and this microfiltrate was used immediately after preparation as a solvent for all components to start in vitro digestions (see below). In this way, only the TiO_2_ that was able to pass the membrane was used, assuring that it can pass the membrane for another time, and any reduction in the migration of the TiO_2_ is related to the presence of the smoothie (reduced “bioacessability”). 

The smoothie (“VICTORIA CYMES”, Wałcz, Poland) was purchased from a commercial outlet in Lublin, Poland. It was composed of water, banana paste (22%), apple juice from juice concentrate (7%), pastes from pears (4.6%) and strawberries (3%), juices from juice concentrates (cherry (1.3%), grape (1.1%) and cranberry (1%)), black carrot concentrate, natural aromas, an acidity regulator (lemon juice), a stabilizer (pectin) and sugar (Appendix A). The product was chosen after preliminary studies of eight similar products (for the full list, engage in personal communication). The studied smoothie was chosen based on its complex composition and the ability to undergo microfiltration in order to mimic “bioaccessibility” during the in vitro digestions. Smoothies that were unable to pass the microfiltration membrane were rejected (no possibility of carrying out the experiment). 

### 2.2. Preparation of the Bacterial Inoculum

*Lactiplantibacillus plantarum* B 4496 was selected based on our previous results, such as a high sensitivity to TiO_2_, including morphological cell changes, suggesting the formation of TiO_2_/bacteria complexes and a possible decrease in the “bioaccessibility” of TiO_2_ [27]. The collection strain (stored at −80 °C) was restored to normal metabolism through triplicate inoculation onto a new MRS medium and culturing (30 °C, 24 h). Post-culture fluids were centrifuged twice (each time to remove the supernatant), and the obtained bacterial biomass was added to the digestions at the “large intestine” stage, as described below.

### 2.3. In Vitro Digestions

The method was adopted from Minekus et al. [29] with only minor modifications, as detailed below. A total of 50 mL of the sample was mixed in the “digestion” chamber (three jacketed glass tanks, each 1 L in volume; Appendix A) with 48 mL of the TiO_2_-containing microfiltrate (produced as described in Section 2.1) and 2 mL of simulated salivary fluid containing 113 mg KCl, 50.3 mg KH_2_PO_4,_ 114.2 mg NaHCO_3_, 3.045 mg MgCl_2_(H_2_O)_6_ and 0.58 mg (NH_4_)_2_CO_3_ [29]. Then, 2 mL of an α-amylase solution containing 451 units of the enzyme (human saliva Type IX-A, 1000–3000 µ/mg protein, Sigma-Aldrich A0521, St. Louis, MO, USA) was added. Next, 0.25 mL of a 0.3 M/L CaCl_2_ solution and 9.75 mL of distilled de-ionized (DDI) water were added, and the “digested” samples were kept at 37 °C for 2 min with stirring at 10 rpm. In the case of each “digested” sample (vessel), a 0.2 µm membrane (VF05P7 PES, Sartorius, France) was connected to the “digestion” vessel with a tube, and the return tube was put back in the tank. The slow working mode was maintained (Cole–Palmer peristaltic pump, approximately 3EN5 mL/min), and 20 mL of the microfiltrate was withdrawn from the “mouth” sample. The microfiltration membrane was used to simulate both “bioaccessibility” and peristaltic movements in the “gastrointestinal tract” model.

Next, Simulated Gastric Fluid (SGF, 60 mL) and pepsin solution (40 mg Sigma P6887 pepsin, dissolved in 6.4 mL SGF) were added, the pH was corrected to 3.0 using an HCl solution (Sigma H1758, for molecular biology) and 0.04 mL of 0.3 M/L CaCl_2_ was added. After 10 min, the second portion of pepsin (40 mg pepsin in 6.4 mL SGF) was added. The volume of all samples was adjusted to 160 mL using DDI water to allow for quantitative analysis. The stirring speed was set to 10 rpm. The samples (20 mL each) were collected using the microfiltration membrane mentioned above after 10, 30, 60 and 90 min (end of the gastric phase). 

Then, 44 mL of Simulated Intestinal Fluid (SIF) was added, followed by the addition of 1.224 g pancreatin (from porcine pancreas, Sigma P1625) and 7.33 g bile salts (Sigma 48305), both suspended in 37.5 mL of SIF. The concentrations of pancreatin and bile salts were adopted from Steward et al. [30]. Next, 0.16 mL of a 0.3 M/L CaCl_2_ solution and 1 M/L NaOH were added to reach pH 7.0, and the “intestinal” volume was complemented to 167.5 mL using DDI water. The stirring speed was set to 10 rpm, and samples (20 mL each) were collected using the microfiltration membrane described above after 0, 40, 80 and 120 min (end of the small intestine phase). 

To start the “large intestine” phase, the inoculum (*L. plantarum*) was added to obtain 10^8^ cfu/g of the digested sample (the addition of inoculum was optimized after optimization experiments; for details, engage in personal communication). The stirring speed was set to 10 rpm, and samples (20 mL each) were collected after 0 min, 10 h and 24 h (end of the large intestine phase). Bacterial plate counts were obtained [31]. The biomass was fixed using a fixative solution (2.5% glutaraldehyde in 0.1 M PBS) for the purposes of microscopic analyses. Simultaneously, a reagent (blank) sample containing the “digestion” fluid in DDI water was run in duplicate in the same manner. This sample was inoculated with lactic acid bacteria and used as the “control”. The content of Ti in the control sample was subtracted from the content in other samples at the corresponding stages of “digestion”. During the whole “digestion”, CO_2_ (analytical grade, Linde Gas Poland, approximately 5–8 mL/min) was run individually through each “digested” sample using a sterile PTFE 0.45 µm syringe filter. Beginning from the “gastric” stage, all additives given to the “digestive” fluid were given using the peristaltic pumps in order to avoid aeration. The samples taken from the GIT were stored at 80 °C until they were analyzed.

### 2.4. Inductively Coupled Plasma Optical Emission Spectrometry (ICP-OES)

The Ti concentration was analyzed using an Inductively Coupled Plasma Optical Emission Spectrometer, ICP–OES (Thermo Scientific iCAP 7200, Waltham, NJ, USA). The axial view was used for metal determinations. The spectral line of 323.452 nm was chosen to obtain the highest sensitivity and minimum interference. The accuracy and precision of the analysis were checked every 10 measurements against the Titanium ICP standard (Centripur®, 12237 Merck KGaA, Darmstadt, Germany). Each sample was measured in triplicate.

### 2.5. Transmission Electron Microscopy (TEM) and Scanning Electron Microscopy (SEM)

The size (morphology) of the nanoparticles was determined using a TEM transmission electron microscope (FEI Tecnai G2 T20 X-Twin Ltd., Tokyo, Japan). The specific methodology and characteristics of the particles are detailed in our previous publication [27].

The technique of scanning electron microscopy (SEM) was employed to visualize changes in bacterial cell morphology. Samples were collected from the “large intestine” section (after adding the bacteria) from three digestive variants: digestive fluid + bacteria (bacterial); digestive fluid + bacteria + TiO_2_ (TiO_2_); and digestive fluid + bacteria + TiO_2_ + smoothie (TiO_2_ + smoothie). The samples were centrifuged at 3000 g for 20 min to obtain pellets. After washing twice with phosphate-buffered saline PBS (0.1 M; pH 7.2,), the pellets were suspended for 2 h at 4 °C in a fixative solution containing 2.5% glutaraldehyde in 0.1 M PBS. Then, the fixed cells were rinsed twice using PBS. Post-fixation was performed for 2 h (4 °C) with freshly prepared 1% OsO_4_. The subsequent rinsing was carried out using a 0.1M phosphate buffer (pH 7.2). Next, the cells were dehydrated in a series of ethanol gradients: 30%, 50%, 70%, 90% and 100% (each time for 10 min). The next step entailed chemical drying with the application of 98% hexamethyldisilazane (HMDS). Eventually, the specimens were coated with gold using an Emitech K550X Sputter Coater. The analysis was performed with a TESCAN vega 3 LMU scanning electron microscope (Brno, Czech Republic) using the secondary electron mode.

### 2.6. Statistical Analysis

Routine statistical tests (means and standard deviations) were performed, and statistical differences (using Tukey’s HSD test) with a significance threshold of *p* < 0.05 were determined using Statistica 13.1 software (StatSoft, Kracow, Poland). 

## 3. Results and Discussion

### 3.1. Transmission Electron Microscopy (TEM) Analysis of the Samples

The distribution of the particle sizes, determined using TEM, revealed that 25.93% of the analyzed TiO_2_ was composed of particles under 100 nm in size. The ζ value ranged from +40 mV (pH = 2) to −17mV (pH = 10). The determined isoelectric point (IEP) was pH = 7.8. The ζ value was positive below these pH values and negative above these values (Figure 1).

### 3.2. Content of Ti in Ultrafiltrates after In Vitro Digestion

The “bioaccessibility” of nutrients in the in vitro gastrointestinal tract model was simulated using the microfiltration membrane (0.2 µm). Prior to digestion, as well as at the “oral cavity” stage, it could be observed that Ti penetrated the ultrafiltration membrane in higher concentrations if no smoothie had been added (Figure 2). We suspect that the addition of the smoothie most likely led to the entrapment of TiO_2_ within the three-dimensional structure characteristic of the polymers present in the smoothie (mainly non-starch polysaccharides such as pectin, cellulose and hemicellulose, as well as polyphenols, whose ability to bind NPs has been well documented in recent decades) [32,33]. As TiO_2_ is, in this case, chemically neutral, we suspect that the oxide did not react chemically with the ingredients of the smoothie but was instead mechanically trapped and therefore could not be “absorbed” in our simple model of “bioaccessibility” utilizing a microfiltration membrane. Importantly, one should note that the TiO_2_ present in the “digested” samples was selected from the bulk commercial E171 by first passing it through the same microfiltration membrane as the one used during the “digestions” (see Section 2.1). This allowed us to ensure that the only fraction of TiO_2_ involved in the digestions was the one not stopped by the membrane. 

It was also demonstrated that the content of the “bioaccessible” Ti (i.e., passing through the ultrafiltration membrane) was significantly influenced by the presence of bacteria. During the large intestine stage, after the addition of the inoculum, we observed a significant decrease in the content of Ti in the microfiltrate in the presence of the food matrix, as opposed to the sample containing digestive fluids but no smoothie. Our assumptions were further confirmed by the results of the plate analysis and SEM imaging (Table 1, Figure 3). The studies revealed that TiO_2_ significantly reduced the growth of the bacteria after 10 h of adding the microbial inoculum (Table 1).

### 3.3. Scanning Electron Microscopy (SEM)

SEM was employed to analyze the morphology of *L. planatrum* (Figure 3). 

The cells were present in the “digested” samples for 0, 10 or 24 h in different combinations. The control cells were rod-shaped bacteria, and the images revealed the presence of typical long, intact and slender thalli bacteria. (Figure 3A–C). SEM imaging after exposure to TiO_2_ was also conducted, and the morphological changes of the cells were clearly visible. In comparison with the control bacterial cells, the topography of some cells was altered, some cells appeared to have shrunk and some had collapsed walls. The topography of other cells revealed small or deeper indentations. Additionally, some cells were covered by clusters of nanoparticles (Figure 3D–F). Lesions and changes were also observed in cells incubated in the presence of TiO_2_ together with the smoothie. Corresponding electron micrographs representative of the above are shown in Figure 3G–L. The changes were more commonly found in comparison with previous micrographs: clear and well-discernible damage, deep indentations in deformed cells and collapsed walls due to the leakage of the contents. A higher affinity of TiO_2_ to the surface of the examined cells was observed because more particles were attached to the cells.

Under the simulated conditions of the gastrointestinal tract, the physicochemical properties of TiO_2_ NPs undergo significant changes, as does the structure of the NPs themselves [34], which may affect their absorption in the small intestine [35] as well as their impact on the intestinal microbiota in the large intestine [36]. The content of salt, the diversity of the food matrix and changes in the pH may all potentially influence the behavior of TiO_2_ during its journey through the gastrointestinal tract. In the course of in vitro digestion, TiO_2_ comes in contact with acid in the stomach (pH 3 or less) as well as complex electrolyte solutions in the presence of various organic fractions (proteins, lipids, etc.), which likely influences its physicochemical properties and determines the fate of TiO_2_ in vitro [37].

The absorption of proteins, lipids and other chyme molecules from food results in the formation of biomolecular coronas that impact the modulation and characteristics of TiO_2_ particles’ surface characteristics [38], which in turn influences their behavior in complex biological systems as well as the relevant cellular/tissular response [39]. Coreas et al. [39] demonstrated that the permanent absorption of chyme biomolecules can take place on the surface of TiO_2_. By simulating three-stage in vitro digestion (oral cavity, stomach and small intestine), the researchers were able to observe that the composition of the corona was different before and after digestion. They demonstrated a high diversity of the biomolecules absorbed on TiO_2_ and a dependence of the corona composition on the composition of the food matrix. The researchers attributed this variability to the fact that the corona composition changed with the passage of chyme during the digestive process, while the accumulation of some lipids in the corona was increased, as they contained functional groups with a high affinity for TiO_2_. Bing et al. [40] analyzed interactions between TiO_2_ NPs and typical plant proteins (glutenin, gliadin, zein and soy protein). They demonstrated that all the proteins interacted with TiO_2_ NPs and formed large complexes composed of nanoparticles surrounded by proteins. Laloux et al. [41] demonstrated the aggregation of nanoparticles in the presence of food matrices and the fact that food components in a simulated human gastrointestinal tract were able to stabilize TiO_2_ in the form of a suspension. The entrapment of TiO_2_ by plant polymers inside a 3-D net is highly likely. Li et al. [3] studied interactions between TiO_2_ NPs and some polyphenols. They demonstrated the considerable significance of the respective polyphenols’ chemical structure, as it determined the affinity of their bonding on the surface of TiO_2_ NPs. The authors observed the highest bonding affinity for polyphenols whose structure included three adjacent hydroxyl groups. Using the simulated gastrointestinal tract, they additionally observed the formation of large aggregates composed of polyphenols and TiO_2_ NPs, which were unable to penetrate the dialysis membrane used to simulate the small intestine (epithelial cells); this suggests that the same thing reduced the bioavailability and, most likely, the bioactivity of TiO_2_ NPs [32]. Similar conclusions were reached by Li et al. [33], whose subsequent study demonstrated that tea polyphenols can bond with TiO_2_ particles and that, depending on the specific polyphenols present in tea (catechins, epigallocatechin gallate—EGCG, gallocatechin gallate—GCG or epicatchin gallate—ECG), this effect could be enhanced. In this case, the strongest effect was observed for GCG. Yuso et al. [42] reported that saccharose and bovine serum albumin reduced the size of the agglomerates and stabilized TiO_2_ NPs [42]. Li et al. [32] found that the solubility of TiO_2_ NPs is strongly influenced by the presence of the food matrix as well as the simulated “digestive” fluids. The researchers reported reduced concentrations of Ti in both the presence and absence of the food matrix, which is similar to the results observed in the present study. Furthermore, they demonstrated, in the presence of the food matrix, a moderately high solubility of TiO_2_ during digestion as compared to analogous samples prior to digestion. In other cases, in the absence of the food matrix and with the exception of the intestinal stage, the researchers reported lowered TiO_2_ NPs solubility [32]. 

In our study, we demonstrated that changes in pH can significantly impact TiO_2_ “bioaccessibility” (measured, in this case, with the use of microfiltration). We observed a significant decrease in Ti content in the “gastric” stage (Figure 2). The reduced “bioaccessibility” may have resulted from the lower pH in this section of the gastrointestinal tract (pH 3), which intensified the agglomeration of nanoparticles and consequently reduced the solubility of NPs in our simulated digestive tract. Then, along with an increase in pH in the "duodenum", we observed an increase in TiO_2_ concentration in the microfiltrate and, thus, increased "bioaccessibility", which suggests the breakdown of agglomerates into smaller forms. The results of previous studies have shown that the stability and aggregation of both food-grade and industrial-grade TiO_2_ NPs depend on the pH of the solution [5]. As demonstrated by [35], pH inside the stomach influences the agglomeration of NPs, as it impacts the particles’ surface charge. Mortensen et al. [43] demonstrated that, as expected, during simulated gastric digestion, E171 TiO_2_ was not dissolved but became more susceptible to aggregation as a result of the simulated process. Similar conclusions were reached by Cho et al. [44], who observed that TiO_2_ did not dissolve or was only negligibly dissolved after 24 h in pH 1.5. 

The gastrointestinal tract is the most heavily colonized organ, containing over 70% of all microorganisms living in our bodies [11]. Our previous studies demonstrated that the addition of TiO_2_ had a negative (toxic) impact on bacteria [10,27], and similar results have also been reported by other authors. Planchon et al. [45] and Radziwił et al. [46] demonstrated, in their respective studies, that some bacteria exposed to TiO_2_ NPs would become completely covered by the oxide, while other parts of the bacterial population remained free of TiO_2_, which can lead to discrepancies in terms of metabolism and proteome. Limage et al. [11] showed, in an in vitro model of the gastrointestinal tract, that the exposure of L. rhamnosus to TiO_2_ triggered changes in the thickness of the mucous layer. The production of mucus in the presence of bacteria was significantly altered due to the exposure to TiO_2_ NPs. Kim et al. [47] observed a lower survival of Lb. acidophilus ATCC 43121 bacteria, even after the strain was protected by encapsulation. Additionally, Ding and Shah [48] observed the inhibition (by approximately 7 log) of Lb. acidophilus growth. 

The introduction of the bacterial strain in the in vitro gastrointestinal model must be justified and can be performed in three ways: using fecal samples (a mixture of strains), a group of selected strains or a single strain. Using fecal samples is not recommended due to differences in the microbiome of practically every person. Similarly, using a set of strains raises a question as to the basis on which these microorganisms were selected. The application of one strain at a time seems to be the best alternative in the presented work, and we used the L. plantarum strain, as it was shown in our previous works that its growth was decreased in the presence of TiO_2_. Interactions of TiO_2_ with bacterial cells (complexation and/or adsorption) were confirmed in this work [27]. Indeed, the inoculation of the “large intestine” with this strain resulted in interactions with TiO_2_ and led to a decrease in the TiO_2_ levels in the microfiltrate (lowered “bioaccessibility”) of TiO_2_.

## 4. Conclusions

In conclusion, it can be posited that the presence of the smoothie in the “digestive” fluid during the in vitro digestion decreased the rate of TiO_2_ passage through the 0.2 µm microfiltration membrane. This phenomenon can be due to a chemical interaction with the ingredients of the smoothie (this is unlikely given the low reactivity of TiO_2_) or the entrapment of TiO_2_ within the food matrix.

Additionally, the TiO_2_ content in the “digestive fluid” was significantly reduced due the presence of the L. plantarum strain at the “large intestine” stage (due to absorption on the bacterial surface). Plate analysis and SEM imaging revealed that TiO_2_ significantly reduced the growth of the bacteria after 10 h of cultivation.

There are still many unanswered questions regarding the impact of the food matrix on the absorption, distribution, metabolism and release of TiO_2_. It is important to better understand the fate of TiO_2_ in the gastrointestinal tract and the transformations occurring there under the influence of nutrients or chyme if we are to accurately determine the actual toxicity of TiO_2_. The matrix can potentially alter the physicochemical properties of TiO_2_, which can significantly influence the level of its absorption. The various effects of digestive processes, the formation of the protein corona and its physicochemical properties and the bioavailability and potentially harmful consequences of TiO_2_ exposure still remain largely unknown.

Our study was a pilot study serving as an introduction to further research aimed at facilitating a deeper understanding of the processes and reactions taking place inside our bodies with a view to potentially better protecting them.

## Figures and Tables

**Figure 1 nutrients-14-03503-f001:**
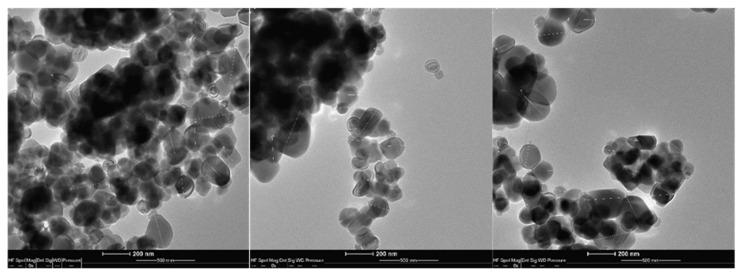
TEM images of the analyzed TiO_2_.

**Figure 2 nutrients-14-03503-f002:**
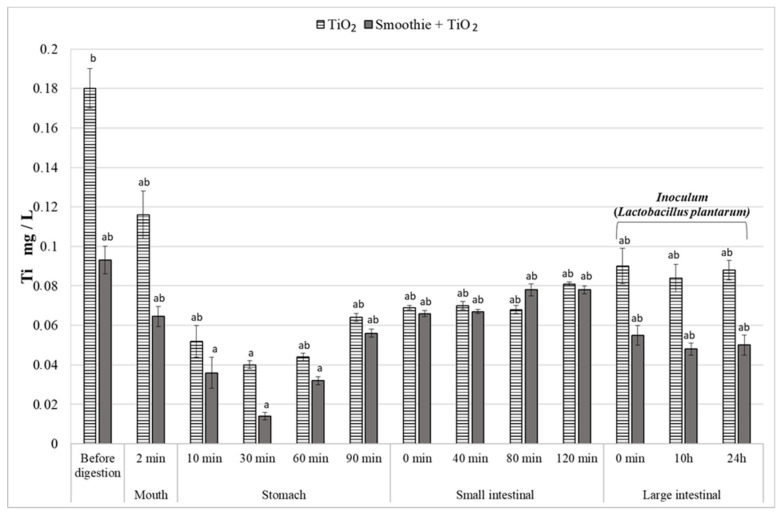
Content of Ti ions (mg/L) at the respective stages of in vitro digestion in the presence of the food matrix (smoothie + TiO_2_) and the digestive liquid alone (TiO_2_), n = 3. Various small letters (a, b) mean significant differences at *p* < 0.05.

**Figure 3 nutrients-14-03503-f003:**
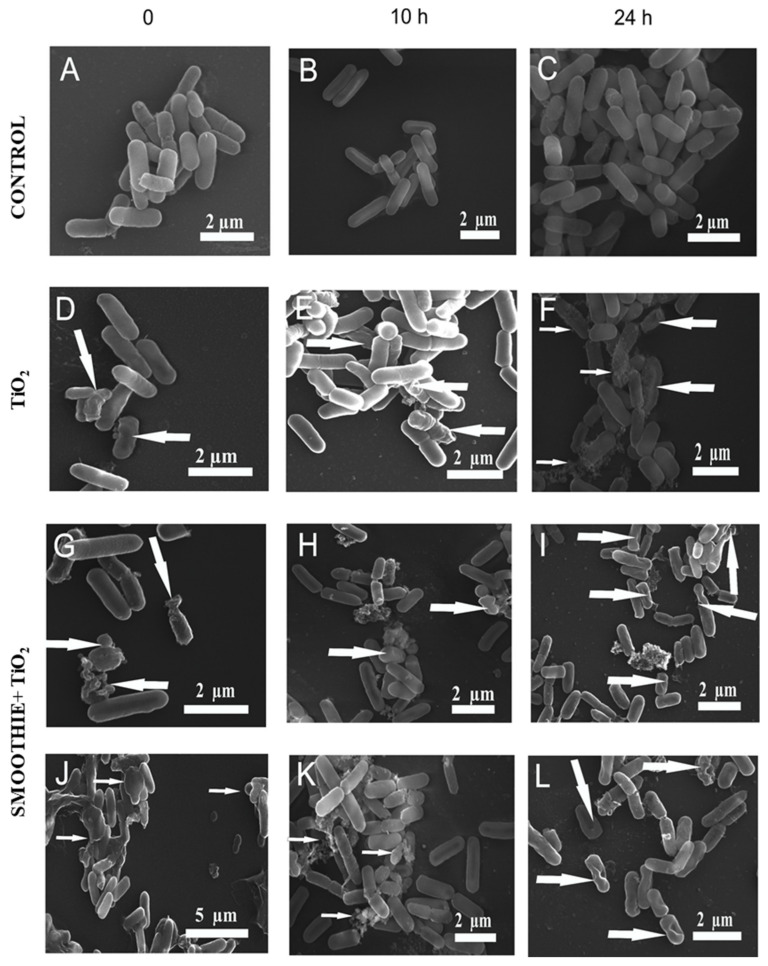
Scanning Electron Microscopy images of Lb. plantarum bacteria cells grown during in vitro digestions (0, 10, 24 h). (**A**–**C**): Control (bacteria only), (**D**–**F**): digestions with TiO_2_, (**G**–**L**): digestions with the smoothie and TiO_2_. Arrows indicate changes observed in cells or nanoparticles. Arrows indicate the deformation of cells or nanoparticles attached to cell surfaces.

**Table 1 nutrients-14-03503-t001:** Inhibition of L. plantarum growth in the in vitro “gastrointestinal tract” model in the presence of TiO_2_.

Digestion Variant	Time from the Addition of Bacteria		
	0 h	After 10 h	After 24 h
	Growth inhibition (%)		
Control (bacterial)	100	17.3	0.29
TiO_2_ + bacterial	100	34.5	1.8
Smoothie + TiO_2_ + bacterial	100	22.1	2.0

## Data Availability

Not applicable.

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
