# Peer review of "Smoothies Reduce the “Bioaccessibility” of TiO2 (E 171) in the Model of the In Vitro Gastrointestinal Tract"

_nutrients, 2022, doi:10.3390/nu14173503_

Round 1
Reviewer 1 Report
This manuscript describes the influence of a smoothie on the bioaccessibility of TiO2. It is important by reinforcing the concept of the influence of a food matrix on the bioaccessibility of substance, i.e. an additive, a drug and more generally any xenobiotic and, consequently, on its bioavailability and potential toxicity. Such an approach, using in vitro digestion and the effect of and on the microbiota should be extended and combined with absorption tests to better approach the bioavailabilty, although considering the reductionist character of a purely in vitro evaluation. In addition to now rather classical in vitro pre-colonic digestion with successive digestive fluids, it includes the use of a colonic bacterial strain, i.e. Lb. plantarum, which although reductionist, provides interesting data on the effect of this alimentary additive on the intestinal microbiota. The authors describe their contribution as “a pilot study” and should be considered as such.
My main, and almost only one, objection concerns the 0,2 µm microfiltration method as a bioaccessibility evaluation. This technique should be better explained and its validation presented since it seems rather unusual.
Fig. 2: the number of samples and repetitions should be indicated in the legend.
No access to the “Supplementary Materials…
Reviewer 2 Report
Authors have presented an interesting study on smoothies and TiO2 bioaccessibility using in vitro digestion model with inoculation of Lactiplantibacillus plantarum B 4496. The manuscript presents data collected from SEM and TEM, TiO2 content in the digesta and inhibition of Lactiplantibacillus plantarum B 4496. The flow of the manuscript must be improved, with removal of information in brackets into actual sentences. Consistency in the use of GIT is required.
Overall, the study presents some important scientific information, however, explanation and discussion should be improved. Often there were not much of justifications over why certain method or certain way of analysis was used. The authors have declared at the end of the manuscript that the work was of pilot study and will need further study.
For further amendments and comments, see below:
Amendments:
Line 2: delete “title”
Revise the title to reflect the content better
Revise the abstract – too short, and not enough information. Aims, methods, results and conclusion should all be present
Line 29: revise. Unclear what the authors are trying to say here
Line 40-46: further elaboration of how 2.5mg NP/kg bodyweight/day translates to in consumption of confectionery by a young child would be appreciated, with serving size of confectionery considered into account.
Line 62: change to GIT
Line70: delete composition and leave what’s in the bracket out of the bracket, as it explains better
Line 78: no need to say food portion. Just call it smoothie throughout the aim
Line 79: change to GIT
Line 86: comment on purity of TiO2
Line 88-89: sonicator settings required
Line 105: what is meant by complex composition?
There are too many of “in brackets” information. Minimise if possible
Section 2.1: was the smoothie purchased once for the study or repeatedly over a period?
Line 135-7 and throughout the in vitro digestion part: how was the SGF and other digestive fluids added? Through peristaltic pump?
Stirring speed used for in vitro digestion seems very slow (10rpm). Is this from a reference? How does it justify against the mixing taking place in the human body?
Line 150: 10 ^ 8. Any justification over why this particular CFU was chosen?
Line 159: what is the justification for sterile CO2 usage here? Not N2 flush for anaerobic environment of the GIT?
Section 2.6: any ANOVA done on data?
Line 267 onwards: what is meant by the term corona and bio-corona in the manuscript?
Line 279-281: not sure whether the quoted study is relevant to the current manuscript. What kind of protein complexes could have formed in the current work? Looking at the smoothie composition, most of the components were carbohydrates.
Line 293-302: what sort of polyphenols were present in what amounts, in the smoothies used for the current study? Looking at the smoothie composition, it seemed like concentrates mixed with water. How compatible is it to compare the polyphenols present in tea by Li et al. to the current study?
Conclusion: what is the justification for using 0.2micron membrane? Does the pore size of 0.2 micrometer represent human membrane pore size?
Round 2
Reviewer 2 Report
Authors have attempted to address the questions and comments raised in the revised manuscript. The revised version seems clearer and more constructive than before.